# Biomechanical Investigation of Head Injuries Caused by Baseball Bat Strikes with Different Bat Sizes and Velocities: A Finite Element Simulation Study

**DOI:** 10.3390/life16010009

**Published:** 2025-12-20

**Authors:** Han Zhang, Jin Yang, Luyi Guo, Jiani Sun, Shangxiao Li, Weiya Hao

**Affiliations:** 1Research Center for Sports Psychology and Biomechanics, China Institute of Sport Science, Beijing 100061, China; zhanghan@ciss.cn (H.Z.); yangj@sus.edu.cn (J.Y.); guoluyi@ciss.cn (L.G.); sunjiani@ciss.cn (J.S.); lishangxiao@ciss.cn (S.L.); 2School of Exercise and Health, Shanghai University of Sport, Shanghai 200438, China

**Keywords:** traumatic brain injury, head injury, blunt injury, injury severity, finite element analysis

## Abstract

Objective: Traumatic brain injury (TBI) represents a significant clinical problem, with the biomechanical mechanisms of striking from different blunt instruments remaining unclear. This study aims to quantitatively evaluate TBI severity under blunt strikes and to assess the effects of strike velocity and blunt instrument size on biomechanical responses to provide a finite element approach for investigating injury mechanisms and informing clinical diagnosis. Methods: A head finite element model incorporating an outer cortical-cancellous-inner cortical bone structure was developed and verified against a previous cadaveric impact study. Strike velocities and blunt instrument parameters, obtained from experiments in which a long bat (LB) and a short bat (SB) were used to strike a dummy head, were applied as the loading conditions in the finite element simulation. Kinetic energy (KE), internal energy (IE), impact force, von Mises stress on skull, intracranial pressure (ICP), and Head_3ms_ acceleration were analyzed as indicators of injury severity. Results: Simulated force and ICP responses agreed with cadaveric experimental data within a 9.8% error. With increasing strike velocity (10–30 m/s), KE, IE, impact force, ICP, and Head_3ms_ all rose, while von Mises stress evolved from localized to dispersed distribution. Head_3ms_ reached an injury threshold of 80 g at a strike velocity of 10 m/s, and ICP peaks for LB and SB exceeded the brain injury threshold (235 kPa, ≈1760 mmHg) at 12 m/s and 14 m/s, respectively. At the same velocity, LB generated higher KE, IE, impact force, ICP and Head_3ms_ than SB. At 30 m/s, LB generated 390 J KE and 29.0 kN peak force, which were 50.0% and 11.1% higher than those of SB (260 J, 26.1 kN). Conclusion: This study reveals that increasing strike velocity and employing a larger blunt instrument elevate biomechanical responses, resulting in von Mises stress transitioning from localized concentration to multipolar dispersion. Specifically, when striking the head with the LB at velocities exceeding 12 m/s or with the SB exceeding 14 m/s, the impacts indicate a severely life-threatening level. These findings deepen our understanding of the mechanisms of blunt TBI. The constructed and validated finite element model can be repeatedly used for computer simulations of TBI under various blunt striking conditions, providing a scientific basis for clinical diagnosis and surgical planning.

## 1. Introduction

Closed-head injuries caused by blunt strikes are among the most frequent causes of traumatic brain injury (TBI) in violent assaults, contact sports, and traffic accidents [1]. According to the U.S. Centers for Disease Control and Prevention (CDC), more than 580 TBI related hospitalizations and 190 deaths occur daily [2], imposing a substantial burden on patients’ quality of life (QoL) and healthcare systems. The craniocerebral region’s complex anatomy and fragile tissues mean that even moderate external forces can cause irreversible damage, including intracranial hemorrhage, parenchymal injury, and skull fracture [3,4]. These outcomes are central concerns in both neurosurgery and forensic biomechanics.

When the applied load exceeds the skull’s structural capacity, the brittle, three-layered cranial bone experiences stress transfer, fractures, and deformation of the brain [5]. Rotational acceleration is widely recognized as a critical determinant of diffuse axonal injury (DAI) [6], frequently resulting in axonal tensile strain, brainstem avulsion, and ventricular deformation. Clinically, these pathologies manifest as impairment of consciousness, vegetative states, or mortality. Previous studies [7] have demonstrated that density gradients within brain tissue facilitate the concentration of shear strains at the interface between deep and superficial layers during rotational motion, significantly elevating the risk of axonal rupture. Consequently, the severity of TBI is strongly related to the impact’s kinematic parameters, including strike velocity, linear and rotational acceleration, and duration [8,9,10]. To quantify biomechanical responses and understand injury mechanisms, several studies have conducted cadaveric head impact experiments. Nahum et al. [11] struck cadaver heads with a rigid impactor, measured ICP at the frontal and occipital sites using parenchymal sensor and found that peak ICP at both sites was non-linearly related to injury severity indices. Ward et al. [12] reported that an ICP level of 173 kPa corresponded to a moderate risk of brain tissue injury and pressures above 235 kPa (≈1760 mmHg) were associated with severe or even fatal damage. Hardy et al. [13] employed a high-speed biplanar X-ray system and neutral density targets to capture brain motion and strain under impact. They demonstrated that rotational acceleration significantly influences tissue deformation. Yoganandan et al. [14] applied dynamic loading to cadaveric skulls using a drop-weight system. They then analyzed fracture patterns and force-displacement relationships via high resolution CT imaging, identifying the critical stress thresholds for skull fracture. Smith et al. [15] simulated high-velocity impacts with a pneumatic device equipped with strain gauges and pressure sensors to evaluate skull strain distribution and ICP fluctuations. Their results indicated that localized strain concentrations strongly correlated with the initiation of fractures. While these studies established ICP and strain as key indicators of craniocerebral injury, the clinical translation of these findings is often limited by the complexity and ethical constraints of cadaveric tests.

However, ethical considerations preclude cadaveric experiments involving direct strikes with handheld instruments. Therefore, finite element (FE) simulations have been adopted as an alternative approach for impact simulation and dynamic reconstruction, offering a means to investigate injury mechanisms. Li et al. [16] used an FE model to analyze skull stress distribution under blunt impacts. They found that cranial sutures, owing to discontinuities in stiffness, acted as stress concentration zones prone to fracture. The simulated fracture patterns closely matched the cadaveric results reported by Yoganandan et al. [14], confirming the reliability of the FE approach. Kleiven et al. [17] further enhanced a comprehensive head model by incorporating nonlinear brain tissue properties, the meninges and cerebrospinal fluid to simulate dynamic brain deformation under rapid impacts. Their results indicated that maximum principal strain (MPS) and peak ICP together determined injury severity, especially under rotational acceleration. Li et al. [18] reconstructed forensic blunt force head injury cases using a fifth-percentile Chinese head FE model. They found that the simulated peak ICP values far exceeded the 235 kPa injury threshold. The predicted locations of brain contusions and skull fractures matched autopsy findings, highlighting the potential of FE methods for quantitatively evaluating blunt head injury severity. Collectively, these studies demonstrate that FE simulations can replicate the biomechanical responses observed in cadaveric tests, while also allowing analyses of injury severity under various impact conditions. Nevertheless, most current studies rely on case specific analyses [19] and oversimplified bone models that neglect the skull’s three-layered structure [20]. Few studies have systematically quantified the influence of strike velocity or instrument size on TBI severity [21], which are critical for developing effective prevention strategies and treatment plans.

Therefore, this study aims to quantitatively evaluate the severity of TBI under blunt strikes by developing an FE head model with a trilaminar cranial bone structure. The model was used to investigate the effects of strike velocity and blunt instrument size on biomechanical responses, providing a scientific basis for improved injury prediction, clinical diagnosis, and the optimization of protective equipment design to ultimately mitigate the long-term impact on patients’ QoL.

## 2. Methods

### 2.1. Striking Experiment

A total of 36 young adult participants were recruited (18 males and 18 females). The male participants (age: 24.4 ± 1.7 years; height: 175.2 ± 6.3 cm; body mass: 77.4 ± 11.8 kg; right arm length: 56.6 ± 3.4 cm) and female participants (age: 24.6 ± 1.7 year; height: 163.7 ± 6.0 cm; body mass: 62.1 ± 9.9 kg; right arm length: 51.7 ± 3.4 cm) were all right-handed and in good physical and mental health. Two sizes of wooden baseball bats were used, the long bat (LB: length 74 cm, mass 727 g, max diameter 5.2 cm, moment of inertia 3.21 × 10^−2^ kg·m^2^) and the short bat (SB: length 52 cm, mass 442 g, max diameter 4.5 cm, moment of inertia 0.88 × 10^−2^ kg·m^2^). Both bats are shown in Figure 1. Kinematic parameters during bat strikes on a dummy were recorded using motion capture system (Oqus700, Qualisys Track Manager, Sweden). More details regarding the experimental setup and data processing can be found in Yang et al. [22] and Yang et al. [23]. The study was conducted in accordance with the ethical principles of the Declaration of Helsinki and approved by the China Institute of Sport Science institutional review board (CISSLA-20230310). All participants signed informed consent after being fully informed about the study procedures and the use of their data.

### 2.2. Finite Element Modeling

A detailed head FE model was developed in HyperMesh (v.2022; Altair, Troy, MI, USA), comprising a three-layer sandwich structure (outer cortical bone, cancellous bone, and inner cortical bone) and brain tissues (gray and white matter) enclosed in CSF (see Figure 2; mesh details in Table 1). Cortical and cancellous bone differ substantially in structure and function; cortical bone is dense, providing mechanical strength and higher yield stress, whereas cancellous bone has a porous trabecular structure that favors energy absorption and weight reduction, resulting in lower strength and yield stress. Material properties (Table 2) were implemented in LS-PrePost (v.971; LSTC, Livermore, CA, USA) based on published data [24,25,26].

Bat models were reconstructed using CAD reverse engineering based on their geometric dimensions and then meshed in HyperMesh (Figure 3). The bat models contained 50,000–80,000 elements. Bats were modeled as isotropic solids with plastic hardening and damage failure criteria to simulate strike deformation. Element erosion was employed with a principal strain failure threshold set at 0.02.

Explicit dynamic analysis was performed using LS-DYNA (v.971). Translational degrees of freedom at the skull base were constrained to replicate experimental support. Bone tissue failure was simulated with MAT_EROSION, elements were eroded when maximum principal stress reached 98 MPa or principal strain exceeded 0.02. Computational stability was ensured via mass scaling (CONTROL_TIMESTEP) and energy conservation control (*DATABASE_ENERGY). Response data were extracted using *DATABASE_HISTORY_NODE and *DATABASE_BINARY_D3PLOT.

Post-processing was performed in LS-PrePost to acquire key biomechanical metrics (stress distribution, kinetic energy evolution, strike load). ICP ≥ 235 kPa was set as the high-risk brain injury threshold [12].

### 2.3. Model Validation

The FEM validation was conducted under conditions consistent with Nahum et al.’s [11] cadaver experiment (Expt. 37). A 5.6 kg impactor struck the skull at 6.3 m/s at a 45° angle to the Frankfort plane, targeting the frontal region. The skull model remained stationary prior to strike and was configured in an unconstrained free state. The three intracranial monitoring points (frontal, parietal, occipital), measured ICP-time curves and frontal impact force-time curves for experimental-simulation comparison are shown in Figure 4.

### 2.4. Striking Simulations

Following FE head model validation, computational striking simulations were conducted. To quantify the biomechanical responses dependent on velocity and size, 22 simulation conditions were established. A total of 22 simulation conditions were established using a bivariate combinatorial design, incorporating 11 strike velocities ranging from 10 to 30 m/s (10, 12, 14, 16, 18, 20, 22, 24, 26, 28, and 30 m/s; denoted V_1_–V_11_) with two blunt instrument sizes (LB and SB). The impact velocities were obtained from the experimental study (see Section 2.1), in which recruited participants performed full-force strikes on anthropomorphic dummy heads. Key biomechanical responses were extracted in LS-PrePost. During the simulation, contact phase commenced when the resultant force between bat and head exceeded 0 N and terminated at complete separation. For consistency, a unified coordinate system aligned with the bat’s center of mass (COM) trajectory was established, and key structural regions (e.g., cranial surface nodes, central brain elements) were sampled as monitoring points. The physical meanings, units and equations are as follows:(1)Peak intracranial pressure (ICP) (kPa)

Maximum instantaneous pressure was recorded at intracranial monitoring points during contact. It reflects the quantified pressure wave amplitude in brain tissue under violent strike. Equation is as follows:
(1)$${ICP}_{max}=maxp(t)$$ where *p*(*t*) = instantaneous pressure at monitoring points (typically biphasic: negative/positive alternation), with the peak defined as the maximum positive value.


(2)Head_3ms_ acceleration (Head_3ms_) (g)


Maximum average acceleration within a 3 ms window centered at the time of peak head acceleration was calculated to evaluate short-duration, high-intensity strike loading.

Peak acceleration timing
$t_{peak}$ was identified on the acceleration-time curve, and then a continuous 3 ms window was extracted symmetrically around it. If edge data was insufficient, it was extended asymmetrically, or the maximum available length was used. Equation is as follows:
(2)$${Head}_{3ms}=\underset{∆t=3ms}{max}\left(\frac{1}{∆t}\int_{t_{peak-\Delta t∕2}}^{t_{peak+\Delta t∕2}}\alpha \left(t\right)dt\right)$$ where
$\alpha \left(t\right)$ = linear acceleration at head’s COM at time *t*; Δ*t* = 3 ms (or the longest available duration when the full 3 ms window cannot be centered at
$t_{peak}$ due to signal boundaries).

## 3. Results

### 3.1. Model Validation

The impact experiment and simulation results reveal that the impact force-time curve rises rapidly to a peak of 9.0 kN at 5.2 ms, followed by decay, yielding a pulse width of approximately 8 ms. In Figure 5, compared to the cadaver experiment peak force of 8.2 kN, the simulation exhibits high consistency in loading rate, wave curve, and duration, with a maximum peak force error of 9.8%.

As shown in Figure 5, ICP responses at frontal, parietal, and occipital monitoring points peaked at 131 kPa, 69 kPa, and −47 kPa, respectively. These aligned with experimental values (140 kPa, 70 kPa, −48 kPa) within errors of 6.9%, 1.4%, and 2.1%. In addition, the characteristic “coup-contrecoup” pressure pattern-positive pressure at the strike frontal site and negative pressure in the contralateral region replicated the ICP propagation reported by Nahum et al. [11].

### 3.2. Energy

As shown in Figure 6, during the initial phase of bat-to-head impact (0–1 ms), the bat exhibited a sharp decline in kinetic energy (KE) and a simultaneous rapid increase in internal energy (IE) under all simulated conditions. Compared with the SB, the LB exhibited lower rates of KE decrease and IE increase, after which both energies stabilized. For both bats, the peak KE and IE increased with rising strike velocity (10–30 m/s). At same velocities, the LB produced higher maximum KE and IE than the SB.

### 3.3. Force

As shown in Figure 7, all strikes completed within 0–2 ms, while LB at 30 m/s took 3 ms, exhibiting unimodal force-time curves. Striking by SB caused higher peak force loading and unloading rates than LB, resulting in steeper curves during both phases. Peak force increased with increasing velocity, and LB consistently exceeded SB at matched velocities. At 30 m/s, LB reached 29.0 kN, while SB was 26.1 kN.

### 3.4. Stress Distributions

As shown in Figure 8, stress levels within the cranial strike region increased with strike velocity, accompanied by progressive expansion of stress concentration areas. LB consistently produced higher stress responses than the SB. As the strike velocity increased from 10 to 30 m/s, the von Mises stress distribution on the skull surface evolved from localized concentration to widespread diffusion. For the LB, the peak von Mises stress increased from approximately 46 MPa at 10 m/s to about 90 MPa at 30 m/s, whereas for the SB, the peak stress increased from approximately 50 MPa to about 105 MPa over the same velocity range. At low velocities, stress concentrations appeared as single circular regions of low magnitude, confined near the impact point. With increasing velocity, stress distribution areas expanded and developed into annular, band-shaped, or bimodal patterns. Stress propagation toward the skull base induced fracture formation under high-energy conditions. At 30 m/s, LB strikes generated linear fractures accompanied by spalling of cortical bone. Compared to LB, SB generated lower peak von Mises stress and more compact stress concentration zones. These stress distributions were primarily localized near the strike site, manifesting as small patches or asymmetric streaks, without multipolar diffusion or spalling features.

### 3.5. ICP

As shown in Figure 9, ICP fluctuation amplitudes increased with increasing velocity. Peak ICP occurred approximately 0.5 ms after impact in all cases, and ICP gradually stabilized after 2.5 ms. Peak ICP values under LB strikes invariably surpassed those of SB, with disparities becoming particularly evident as velocity increased. The 235 kPa threshold was exceeded at 12 m/s (LB) and 14 m/s (SB), respectively.

### 3.6. Head_3ms_

As shown in Figure 10, Head_3ms_ acceleration increased with increasing velocity, and values in LB trials consistently exceeded those in SB trials. For both bat sizes, Head_3ms_ exceeded 100 g at velocities of 10 m/s and above. Head_3ms_ acceleration reached 150–360 g (LB) and 130–300 g (SB) as velocity increased from 10 to 30 m/s. The difference between LB and SB remained stable at 40–60 g.

## 4. Discussion

The quantitative assessment of injury severity is essential for translating biomechanical responses into practical safety metrics and clinically relevant insights. In our study, increasing strike velocity elevated impact force, kinetic energy, internal energy, and Head_3ms_ acceleration, establishing a clear biomechanical basis for head injury.

Higher strike velocities and kinetic energy increase local cranial stress, which elevates both the risk and severity of TBI. In our results, rising strike velocity caused the von Mises stress to shift from a localized concentration to a multipolar distribution. The LB consistently produced higher stress levels than the SB. Previous studies have shown that regions with the highest principal stress are most prone to fracture [27]. This evidence suggests that strikes from an LB are more likely to cause severe cranial damage. The difference can be attributed to the inertial characteristics of the blunt instruments. In our study, the LB had a mass, moment of inertia, and length that were 75%, 265%, and 42% greater than those of the SB. These differences resulted in greater kinetic energy input and more widespread stress propagation, consequently elevating the risk of skull fracture and TBI. These results are comparable to the simulation results by Lindgren et al. [28] demonstrating through simulations that impactor mass influences cranial stress distribution and can even alter fracture pathways. Rashid et al. [29] further reported that heavier blunt instruments generate higher kinetic energy and produce more extensive structural damage, including a greater tendency for bone collapse. In our study, LB strikes at 30 m/s resulted in both linear and spalling fractures. This finding is consistent with Li et al. [18], who showed that a square wooden stick striking the head at 24–30 m/s produced linear fractures. However, Li et al. [18] used a strain-based failure criterion with element deletion, which led to linear and point-specific fracture lines. In contrast, our erosion-based failure criterion integrated both stress and strain thresholds, yielding fracture patterns that more closely resemble real-world injury conditions, thereby enhancing the model’s utility for predicting specific fracture types encountered in clinical practice. In addition, under comparable impact velocities (8–13 m/s), the peak impact force observed in this study (approximately 16 kN) exceeded the maximum contact force reported by Li et al. [16] (approximately 10 kN). This difference may be attributed to variations in blunt instrument geometry, skull modeling strategy, and material property definitions. In particular, the bat used in this study was longer (this study: 52 cm; Li: 25 cm), and a layered skull model was adopted to better represent the mechanical contrast between cortical and cancellous bone, leading to more realistic stress propagation. Additionally, the lack of detailed material parameter reporting in Li et al. [16] may have contributed to discrepancies in impact force responses.

The ICP results further demonstrated that pressure fluctuations intensified as the strike velocity increased. Ward and Thompson [12] reported that peak ICP values above 173 kPa indicate a moderate injury risk, whereas values exceeding 235 kPa may lead to severe or fatal outcomes. In our simulations, ICP rose steadily with increasing velocity, surpassing the 235 kPa threshold at 12 m/s for LB and 14 m/s for SB, indicating potentially fatal injury levels. These critical velocity thresholds serve as quantitative benchmarks for forensic reconstruction and protective gear design. Specifically, identifying these transition points informs the development of helmet standards, suggesting that effective protective equipment must possess energy absorption capabilities sufficient to maintain ICP below 235 kPa even at these moderate impact velocities [30]. This is consistent with a previous study that simulated blunt head strikes with a tee-ball bat and found that the peak ICP tripled as strike velocity increased from 12 to 20 m/s [31]. Gao et al. [32], using a finite element model to simulate punches (effective mass: 1 kg) to the frontal and temporal regions, reported that the peak ICP rose with speed, reaching approximately 170–200 kPa at 10 m/s. This value is close to the ICP produced by SB at 10 m/s in our study, suggesting that instruments with comparable effective mass and velocity can induce similar injury risks. Contact area may also influence head injury severity. Yoganandan et al. [14] reported that reducing the impactor size by half doubled the probability of skull failure.

Further analysis showed that ICP waves propagated rapidly across the cranial cavity, forming a positive pressure peak at the impact site followed immediately by a large negative pressure region. This biphasic pattern matches the classical coup–contrecoup mechanism described by Nahum et al. [11], where pressure reflections and tissue rebound occur after initial compression. Similar pressure patterns were reported in the FE reconstructions of Post et al. [33] and Pavan et al. [34], confirming that this mechanism contributes to both focal and diffuse injuries. However, the magnitude of the negative pressure region produced by both LB and SB strikes in our study exceeded the thresholds reported by Post et al. [33], implying a higher potential for axonal injury and highlighting a previously underappreciated mechanism that may often poor long-term prognosis in TBI patients. In summary, the dynamic ICP patterns quantified for the two bat sizes demonstrate a clear relationship between strike velocity and injury severity thresholds. These results provide a biomechanical basis for evaluating the risks associated with blunt injury and offer practical guidance for the development of protective equipment and enhanced forensic and clinical diagnosis of head injury mechanisms.

Our results indicate that Head_3ms_ values exceeded 100 g across all strike sizes at velocities ≥10 m/s. According to the GM/RT2100 standard by Rail Safety and Standards Board [35], Head_3ms_ > 100 g signifies high risk of severe head injury, highlighting the non-negligible injury potential even at medium-low velocities. Head impact injury is closely related to cranial structure; the human cranium lacks sufficient structural damping against high-energy blunt trauma [36]. The present biomechanical finding confirms that external protective equipment is essential to prevent such severe head injuries under violent impact conditions, directly supporting preventive strategies aimed at preserving neurological function and QoL.

There are limitations associated with this study. Firstly, the selected blunt instruments (long bat and short bat) were chosen to represent typical baseball bats in terms of both length and mass distribution, but they cannot cover all possible blunt instrument sizes that might be used in real-world scenarios (e.g., bricks). Secondly, the striking velocities used in the simulation were obtained from participants striking a dummy head with maximum voluntary effort in the striking experiment, whereas blunt force assaults often involve low-velocity strikes. Thus, future studies should prioritize the establishment of biomechanical injury severities at lower velocities (<10 m/s). Such investigations are critical for defining the transition from mild concussion to structural failure, which would substantially improve the forensic assessment of common assault-related head injuries.

## 5. Conclusions

This study reveals that increasing strike velocity and employing a larger blunt instrument elevate biomechanical responses, resulting in von Mises stress transitioning from localized concentration to multipolar dispersion. Specifically, when striking the head with the LB at velocities exceeding 12 m/s or with the SB exceeding 14 m/s, the impacts indicate a severely life-threatening level. These findings deepen our understanding of the mechanisms of blunt TBI. The constructed and validated finite element model can be repeatedly used for computer simulations of TBI under various blunt striking conditions, providing a scientific basis for clinical diagnosis and surgical planning.

## Figures and Tables

**Figure 1 life-16-00009-f001:**
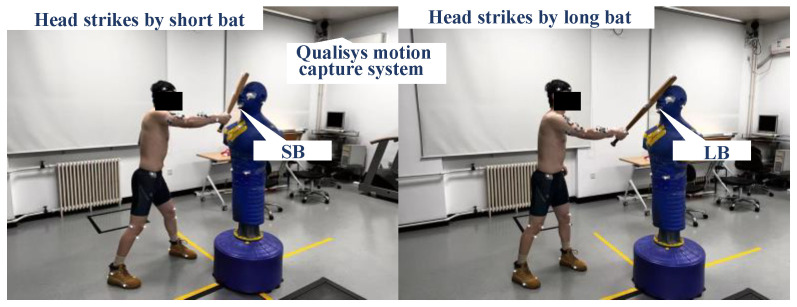
Experimental setup for strikes to a dummy head using a short bat (SB) and long bat (LB).

**Figure 2 life-16-00009-f002:**
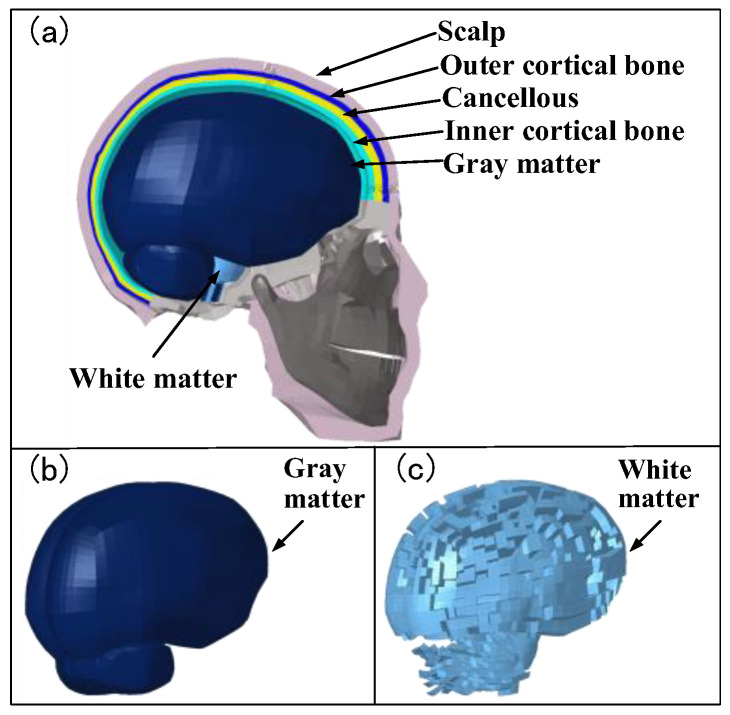
Human head finite element model. (**a**) Cross-sectional view of the head model; (**b**) Gray matter; (**c**) White matter.

**Figure 3 life-16-00009-f003:**
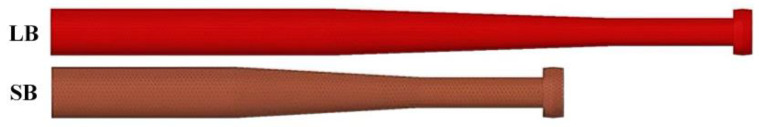
Finite element model of the long bat (LB) and short bat (SB).

**Figure 4 life-16-00009-f004:**
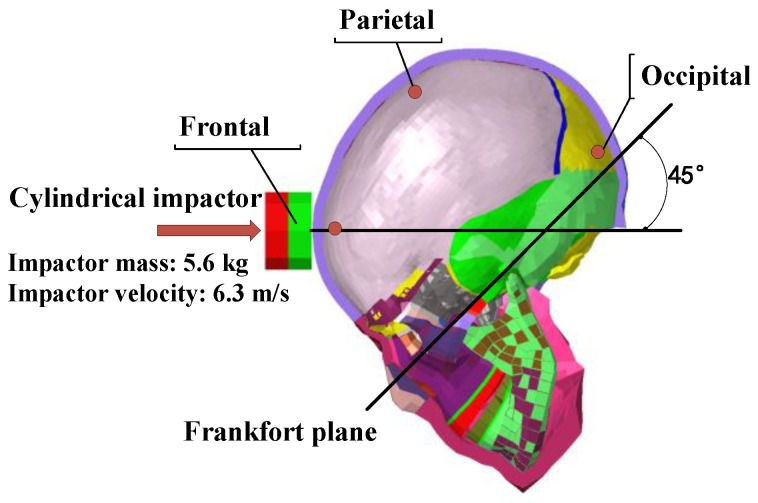
Finite element simulation setup for skull strike based on Nahum et al.’s cadaver experiment [11].

**Figure 5 life-16-00009-f005:**
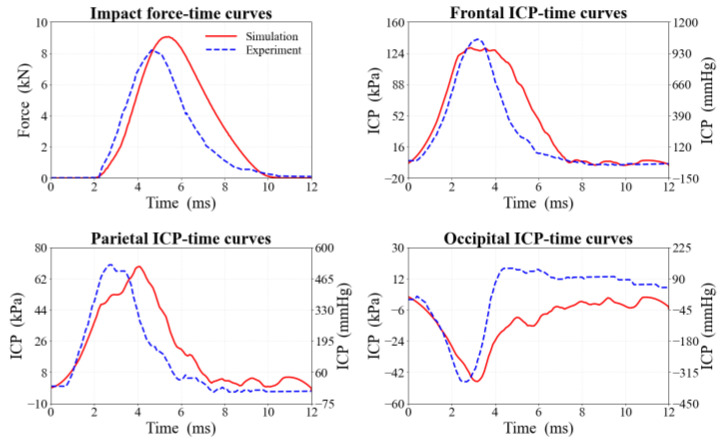
Comparison of impact forces and ICP in different craniocerebral regions between experiment and FE simulation. The right y-axis represents the equivalent values in mmHg (1 kPa ≈ 7.5 mmHg).

**Figure 6 life-16-00009-f006:**
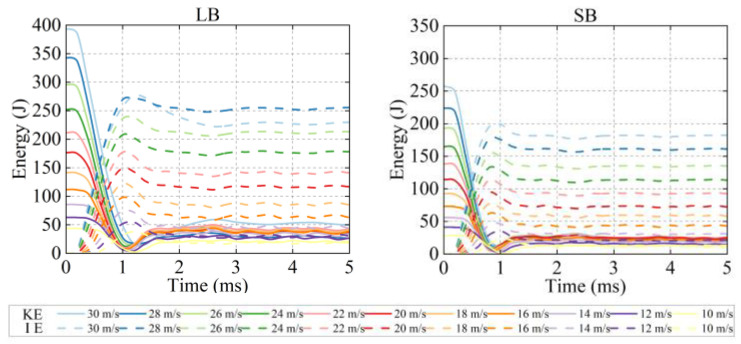
Kinetic energy-time and internal energy-time curves caused by bat strikes at different velocities.

**Figure 7 life-16-00009-f007:**
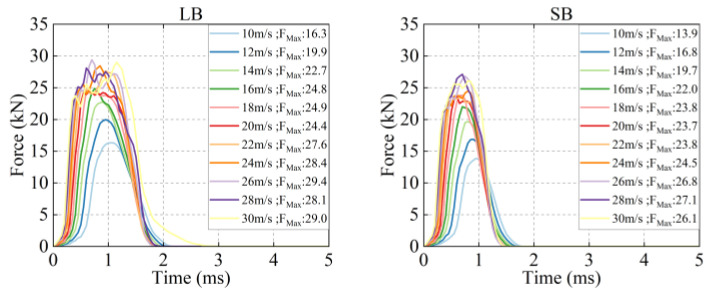
Force–time curves caused by bat strikes at different velocities.

**Figure 8 life-16-00009-f008:**
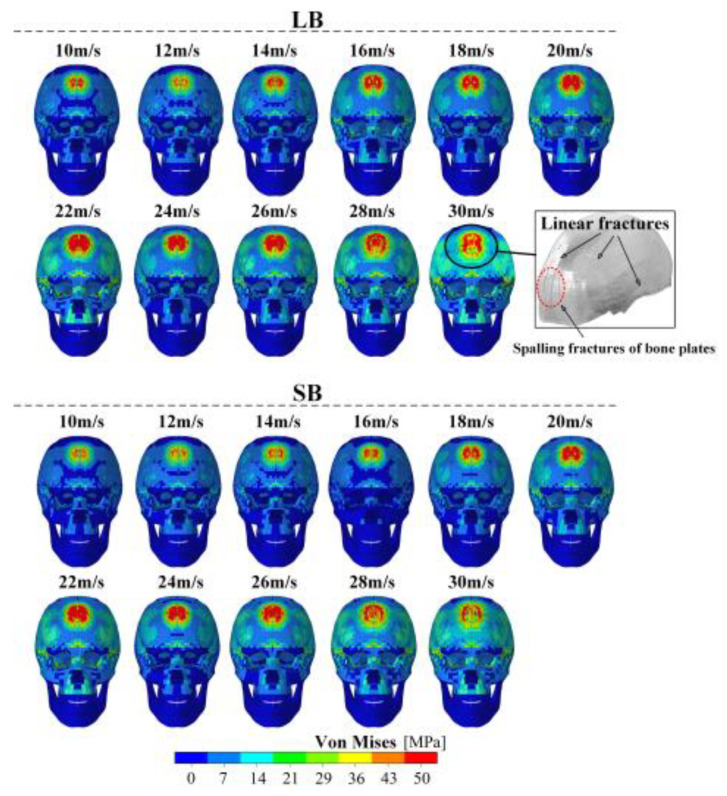
Stress distributions caused by bat strikes at different velocities.

**Figure 9 life-16-00009-f009:**
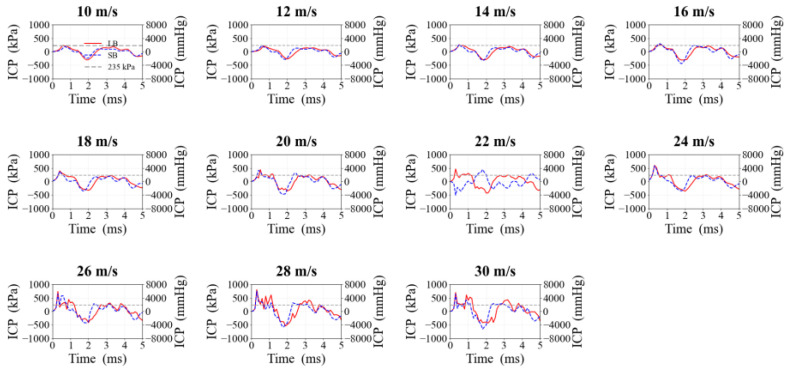
ICP–time curves caused by bat strikes at different velocities. The right y-axis represents the equivalent values in mmHg (1 kPa ≈ 7.5 mmHg).

**Figure 10 life-16-00009-f010:**
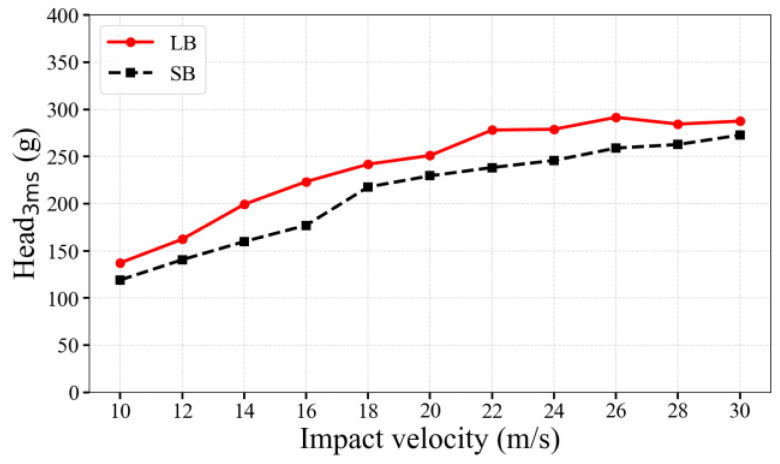
Head_3ms_ acceleration -time curves caused by bat strikes at different velocities.

**Table 1 life-16-00009-t001:** Mesh characteristics and skull thickness of the head finite element model.

Tissue	Thickness	Element Type	Number of Meshes
Outer cortical bone	2.4 mm	Hexahedron	12,654
Cancellous	3.2 mm	Hexahedron	16,872
Inner cortical bone	2.4 mm	Hexahedron	29,526
Gray matter	--	Hexahedron	52,276
White matter	--	Hexahedron	67,564

--: indicates no data available/not applicable/not measured.

**Table 2 life-16-00009-t002:** Material properties and constitutive parameters of head tissues and bats.

Part	Material Model	Density (kg/m^3^)	Poisson’s Ratio	Parameters
Outer cortical bone	Elastic plastic	2100	0.25	E = 15,000 MPa, σ_γ_ = 98.98 MPa
Cancellous	Elastic plastic	1000	0.22	E = 13,700 MPa, σ_γ_ = 5 MPa
Inner cortical bone	Elastic plastic	2100	0.25	E = 13,700 MPa, σ_γ_ = 98.98 MPa
Cerebrospinal fluid	Viscoelastic	1000	--	K = 2000 MPa, G0 = 0.0005 MPa; GI = 0.0001 MPa; λ = 80
Gray matter	Viscoelastic	1000	--	K = 2160 MPa, G0 = 0.006 MPa; GI = 0.012 MPa; λ = 80
White matter	Viscoelastic	1000	--	K = 2160 MPa, G0 = 0.006 MPa; GI = 0.012 MPa; λ = 80
Wood bat	Elastic plastic	600	0.3	E = 10,000 MPa, σ_γ_ = 30 MPa

E: Elastic Modulus; σ_γ_: Yield Stress; K: Bulk Modulus; G0: Initial Shear Modulus; GI: Infinite Shear Modulus; λ: Attenuation Constant; --: indicates no data available/not applicable/not measured.

## Data Availability

The original contributions presented in this study are included in the article. Further inquiries can be directed to the corresponding author.

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
