# Peer review of "Biomechanical Investigation of Head Injuries Caused by Baseball Bat Strikes with Different Bat Sizes and Velocities: A Finite Element Simulation Study"

_life, 2025, doi:10.3390/life16010009_

Round 1
Reviewer 1 Report
Comments and Suggestions for Authors
Dear Authors,
I was asked to review an article titled “Biomechanical Investigation of Head Injuries Caused by Baseball Bat Strikes with Different Bat Sizes and Velocities: A Finite Element Simulation Study”.
The study quantitatively evaluates the blunt strike TBI severity and assesses the effects of strike velocity and blunt tool size on biomechanical responses to injury using a head finite element model, which could assist in computer simulations of TBI.
Although a few studies have systematically quantified the influence of strike velocity or tool size on TBI severity, which represents the strength of the actual study, I have some objections regarding this paper, which I have listed below:
- Highlighting key take-home messages would enhance readability.
- The eligibility criteria should be more properly defined in the Methods.
- It is advised to avoid starting the sentence numerically, which is not grammatically correct (Page 3; Line 106).
- In the Results, besides defining the ICP values in kPa, it would be interesting to also define them in mmHg for easier readability and comprehension.
- In the Discussion, potential sources of bias should be described.
- All References should be cited using standardized abbreviations.
Reviewer 2 Report
Comments and Suggestions for Authors
Dear Authors
Thank you for your professional work. I enjoyed your work. I think your study was very interesting, although I have some comments here that I believe can improve your work.
Overall Summary
This is a well-designed and scientifically rigorous study that makes a meaningful contribution to the field of traumatic brain injury biomechanics. The finite element model is appropriately validated, and the systematic examination of bat size and strike velocity provides novel insights into injury mechanisms. The manuscript is suitable for publication after addressing the following points.
Major Strengths
1. Innovative Approach: The combination of a trilaminar skull model with variable blunt instrument sizes and velocities is a clear advance over previous studies.
2. Strong Validation: Comparison with Nahum et al.’s cadaveric data lends credibility to the simulation results.
3. Clinical Relevance: The findings have clear implications for forensic analysis, protective equipment design, and clinical TBI assessment.
Specific Suggestions for Revision
1. Introduction & Background
Consider briefly introducing the role of rotational acceleration in TBI early in the introduction, as it is discussed later in relation to prior work (e.g., Kleiven et al.).
The transition from cadaveric studies to FE modeling could be slightly smoother—consider a sentence linking the ethical/logistical limitations of cadaver tests directly to the rationale for FE simulation.
2. Methods Section
Material Properties: Please briefly justify the yield stress values for cancellous vs. cortical bone (Table 2), especially the large difference (5 MPa vs. ~99 MPa). A citation or short explanation would be helpful.
Bat Failure Criterion: Explicitly state the failure model used for the bat (e.g., MAT_ADD_EROSION with specific parameters) to enhance reproducibility.
Ethics Statement: Confirm that participant consent was obtained for the striking experiment, even if implied by IRB approval.
3. Results
Figure 8 (Stress Distributions): This figure is referenced but not visible in the submitted draft. Ensure it is included and clearly labeled in the final version.
Quantitative Stress Data: Consider adding a table or summary values (e.g., peak von Mises stress at key velocities) to complement the descriptive analysis in Section 3.4.
Figure 7 Legend: The SB legend appears truncated in the text—please verify completeness in the final figure.
4. Discussion
Low-Velocity Impacts: The limitation regarding low-velocity strikes (common in assaults) is noted but could be expanded. Consider suggesting future work to explore the biomechanical threshold for injury at lower velocities.
Woodpecker Analogy: While interesting, this analogy may distract from the human-focused implications. Consider shortening or reframing it to emphasize human anatomical vulnerability and the need for protection.
Clinical Translation: Elaborate briefly on how the ICP thresholds (12 m/s for LB, 14 m/s for SB) could inform real-world injury assessment or helmet standards.
5. Language & Clarity
The English is generally clear but would benefit from proofreading to improve flow and precision. Examples:
· “Bat models in Figure 3 were reconstructed via CAD reverse engineering according to geometric sizes…”
→ Suggested: “Bat models were reconstructed using CAD reverse engineering based on their geometric dimensions…”
· “Stress propagation toward the skull base induced fractures formation…”
→ Suggested: “Stress propagation toward the skull base induced fracture formation…”
Minor typographical errors (e.g., “Lijc” on page 12) should be corrected.
6. References & Formatting
Ensure consistent journal title formatting (some are italicized, others not).
Verify that all in-text citations correspond to the reference list.
Recommendation
Minor Revisions Required.
The manuscript is scientifically sound, well-structured, and provides valuable insights. With the above revisions, it will be ready for publication and will serve as a useful resource for researchers and clinicians in the field of neurotrauma biomechanics.
Thank you for your professional manuscript
Comments on the Quality of English Language
I think the English in this study is fine, although I have some comments here that I believe can improve your English quality. Below are examples and suggestions for revision.
Areas for Improvement
1. Awkward or Ambiguous Phrasing
· Example:
“Bat models in Figure 3 were reconstructed via CAD reverse engineering according to geometric sizes, and meshed in HyperMesh.”
Suggested revision:
“Bat models were reconstructed using CAD reverse engineering based on their geometric dimensions and then meshed in HyperMesh.”
· Example:
“Stress propagation toward the skull base induced fractures formation under high-energy conditions.”
Suggested revision:
“Stress propagation toward the skull base induced fracture formation under high-energy conditions.”
2. Sentence Structure and Flow
· Example (Introduction):
“Due to the complex anatomy and fragile tissues of the craniocerebral region, even moderate external forces may produce irreversible consequences such as intracranial hemorrhage, parenchymal damage, and skull fracture [3,4], which are central concerns in neurosurgery and forensic biomechanics.”
Suggested revision:
“The craniocerebral region’s complex anatomy and fragile tissues mean that even moderate external forces can cause irreversible damage, including intracranial hemorrhage, parenchymal injury, and skull fracture [3,4]. These outcomes are central concerns in both neurosurgery and forensic biomechanics.”
3. Minor Grammatical and Typographical Issues
· Inconsistent journal header: “Lije” / “Lijc” instead of “Life” (pages 2, 5, 9, 12).
· Article usage:
“a 5.6 kg impactor struck the skull at 6.3 m/s with a 45° angle…”
“at a 45° angle…”
4. Clarity in Technical Descriptions
· Example (Methods):
“Explicit dynamics were solved using LS-DYNA (v.971).”
Clearer:
“Explicit dynamic analysis was performed using LS-DYNA (v.971).”
· Example (Results):
“Head₃ₘₛ exceeded the 80 g injury threshold at 10 m/s…”
More precise:
“Head₃ₘₛ exceeded the injury threshold of 80 g at a strike velocity of 10 m/s…”
Recommendation
I recommend a thorough proofread by a native English speaker or a professional language editing service before final submission. This will enhance the manuscript’s readability and ensure that the high-quality science is presented with matching linguistic clarity.
Thanks
